# *Prenanthes purpurea* and 3,5-DiCQA Alleviate Cellular Stress in H_2_O_2_-Induced Neurotoxicity: An In Vitro Comparative Study

**DOI:** 10.3390/ijms25189805

**Published:** 2024-09-11

**Authors:** Rositsa Mihaylova, Dimitrina Zheleva-Dimitrova, Viktoria Elincheva, Reneta Gevrenova, Georgi Momekov, Rumyana Simeonova

**Affiliations:** 1Department “Pharmacology, Pharmacotherapy and Toxicology”, Faculty of Pharmacy, Medical University of Sofia, 1000 Sofia, Bulgaria; rmihaylova@pharmfac.mu-sofia.bg (R.M.); viktoriaelincheva@gmail.com (V.E.); gmomekov@pharmfac.mu-sofia.bg (G.M.); 2Department of Pharmacognosy, Faculty of Pharmacy, Medical University of Sofia, 1000 Sofia, Bulgaria; dzheleva@pharmfac.mu-sofia.bg (D.Z.-D.); rgevrenova@pharmfac.mu-sofia.bg (R.G.)

**Keywords:** neuroprotection, neurotoxicity, oxidative stress, *Prenanthes purpurea*, polyphenols, DiCQA

## Abstract

Oxidative stress exerts multiple disruptive effects on cellular morphology and function and is a major detriment to age-related and pathological neurodegenerative processes. The present study introduces an evaluative and comparative investigation of the antioxidant and cytoprotective properties of a *Prenanthes purpurea* extract and its major constituent 3,5-dicaffeoylquinic acid (DiCQA) in an in vitro model of H_2_O_2_-induced neurotoxicity. Using validated in vitro and in silico approaches, we established the presence and concentration dynamics of cellular protection in a 24 h pretreatment regimen with the natural products. The conducted cytotoxicity studies and the automated Chou–Talalay analysis for studying drug interactions demonstrated a strong antagonistic effect of the tested substances against oxidative stimuli in an “on demand” manner, prevailing at the higher end of the concentration range. These findings were further supported by the proteomic characterization of the treatment samples, accounting for a more distinct neuroprotection provided by the pure polyphenol 3,5-DiCQA.

## 1. Introduction

Oxidative stress and inflammatory responses play a prominent role in human pathology and are a characteristic trait of a wide variety of non-communicable diseases and conditions, including cancer, atherosclerosis, diabetes, and neurodegenerative and cardiovascular disorders. In the wake of modern socio-economic conditions and lifestyle factors, neurodegenerative disorders emerge as a significant public health concern, representing a large unmet medical need lacking effective treatments [1,2,3,4,5]. Their pathophysiology involves a complex interaction of both general and disease-specific factors that ultimately result in neuronal degradation and apoptosis, leading to cognitive impairment and clinical symptoms [6]. 

Oxidative stress has long been identified as a key general factor in the development of common conditions like Alzheimer’s disease (AD) and Parkinson’s disease (PD). Many aging theories propose that ongoing oxidative damage induces mitochondrial DNA mutations and dysfunction, resulting in metabolic alterations, neuronal senescence, and cell death [7,8,9,10]. A major predisposition for the occurrence and development of oxidative events is the imbalance between oxidants and antioxidants, typically caused by an excess production of reactive oxygen species (ROS) or a malfunctioning antioxidant defense system [4,7,11]. ROS are continuously produced in aerobic organisms as by-products of normal oxygen metabolism [12]. Within physiological ranges, ROS have been shown to act as important signaling molecules in various redox-sensitive pathways, including cell proliferation, apoptosis, and proinflammatory and immunological responses without triggering oxidative damage. However, both endogenous sources, such as the mitochondria, and exogenous causes, such as chemicals and radiation, continuously introduce free radicals into the cellular environment, triggering a switch in ROS function [13,14,15,16].

The brain utilizes a substantially large amount of oxygen for its activities and can therefore be viewed as a major source of free radicals and reactive oxygen species (ROS), making it a central hub for neurodegeneration [11,17]. The ROS involved in neurodegeneration include free radicals like the superoxide anion (O_2_^−^) and hydroxyl radical (OH⁻), as well as non-radical molecules such as hydrogen peroxide (H_2_O_2_) [13,18]. Additionally, DNA, RNA, lipids, proteins, and other macromolecules are also targeted by reactive nitrogen species (RNS) like nitric oxide (NO) and its secondary product peroxynitrite (ONOO−), which are suggested to participate in deleterious inflammatory and degenerative processes resulting in cell death through apoptosis or necrosis [13,17]. Furthermore, ROS overload and consequent oxidative damage appear to be involved in a complex interplay with inflammatory stress responses, known as another major driver in the development and progression of neurodegenerative diseases [19,20,21].

Significant advances have been made in understanding the biochemical mechanisms underlying plant-polyphenol-related antioxidant effects [22]. Species from the Asteraceae family provide a rich source of essential oils, phenolic acids, flavonoids, and sesquiterpene lactones, but the unique nature of their health-promoting benefits, including increased endogenous antioxidant production, has been associated with naturally high levels of acylquinic acids [23,24].

*Prenanthes purpurea* L. (Cichorieae tribe, Asteraceae family) is distributed in Eastern and Southwestern Europe, the Caucasus, and Western Asia (http://www.worldfloraonline.org/taxon/wfo-0000049845 Accessed on 10 July 2024). It grows primarily in the temperate biome. A previous study on the aerial parts of *P. purpurea* revealed the presence of more than 100 specialized metabolites, which were dereplicated/annotated by ultra-high-performance liquid chromatography–high-resolution mass spectrometry [25]. The methanol–aqueous extract from aerial parts showed antioxidant capacity in radical scavenging, reducing power, and metal chelating assays. The high antioxidant activity is related to the phenolic and flavonoid content and a number of caffeoyl conjugates characterized in the samples [25]. The polyphenolic profiling described in the aforementioned study revealed the occurrence of chemical markers such as acylquinnic acids (chlorogenic, 3,4-dicaffeoylquinic, 3,5-dicaffeoylquinic, and 5-feruloyl-2-hydroxyquinic acid) and acyltartaric acids (caffeoyl-dihydro-caffeoyltartaric, caffeoyltartaric, and cichoric acid), together with the hexuronides of eryodictiol, luteolin, apigenin, and naringenin. Thus, the species is considered a rich source of acylquinic acids. In addition to evoking an antioxidant response, *P. purpurea* extract also displays in vitro hepatoprotective effect in diclofenac-induced liver injury, which has generated further interest in the species as potential candidate for antioxidant cellular protection.

It is worth noting that the compounds in the acylquinic acid family are abundant polyphenols in plants [23,26]. Indeed, caffeoylquinic (chlorogenic) acids actively eliminate free radicals and inhibit oxidative injuries and apoptosis in multiple tissues by suppressing caspase activity and triggering pivotal antioxidants, thus scavenging excessive cellular free radicals. As highlighted in recent review articles, the beneficial effects of chlorogenic acid have been demonstrated in a number of experimental models, with the evidence strongly suggesting that it raises antioxidants by activating the nuclear factor erythroid 2-related factor (Nrf2) pathway [23,27]. Chlorogenic acids attenuate the spike in malondialdehyde and ROS levels caused by oxidative stress in different pathological models [28,29,30].

In line with these published research findings, in a previous study, we demonstrated the in vitro antioxidant, antiapoptotic, and cytoprotective activities of the *P. purpurea* species as a pretreatment remedy for drug-induced hepatotoxicity [25].

The present study was intended to further characterize the cytoprotective properties of a *P. purpurea* leaf extract (PE) in an in vitro model of H_2_O_2_-induced neurotoxicity in a comparative manner to 3,5-DiCQA as its major constituent. Using validated in vitro and in silico approaches based on the Chou–Talalay methodology for studying drug interactions, we established the quantitative aspects of neuroprotection against oxidative stress in a 24 h pretreatment regimen with phytoprotectants. A highly specific membrane-based sandwich immunoassay enabled us to track changes in the expression levels and activation status of key proteins related to cell stress and provided further elaboration on the molecular mechanisms of cytoprotection in response to H_2_O_2_-induced cellular injury. 

## 2. Results and Discussion

### 2.1. In Vitro Evaluation of Neuroprotective Activity

#### 2.1.1. Results of Cell Viability Assays

The preliminary evaluation of the neuroprotective effects of the phyto-polyphenol 3,5-DiCQA and the methanol extract of the aerial parts of *P. purpurea* was conducted in an in vitro model of H_2_O_2_-induced neurotoxicity and oxidative stress in human neuroblastoma SH-SY5Y cells. The main objective of the primary cell viability study was to compare the cytotoxic potential of a 30 min exposure to five serial concentrations of the oxidizing agent H_2_O_2_ in a pretreatment and non-pretreatment regimen in a fixed 5:1 ratio. In the pretreated groups, the induction of neurotoxicity was preceded by a 24 h incubation with the test compound 3,5-DiCQA and the natural extract PE.

The results of the conducted cytotoxicity assays definitively showed that the tested substances exerted a stimulatory effect on SH-SY5Y cell survival and proliferation specifically under stressful conditions. The observed neuroprotection appeared to be strongly dose-dependent (especially pronounced in the higher concentration range) and more noticeable in the DiCQA-pretreated cell cultures (Figure 1). Both PE and DiCQA produced their strongest cytoprotective effects beyond the half-inhibitory concentration of the single H_2_O_2_ agent (ca. 500 µg/mL), indicating their “on-demand” action in response to potent damaging stimuli affecting the survival of over 50% of the cell population. As a result, in the PE and DiCQA pretreatment regimens, we found drastic 4- and 14-fold increases in the calculated IC_50_ values of the toxicant, respectively (Table 1).

#### 2.1.2. Results of Compusyn^®^ Analysis

To more thoroughly evaluate the cumulative effects of both pretreatment regimens on cell viability, we subjected the MTT-derived cytotoxicity data to an extended automated analysis in the Compusyn^®^ software (version 1.0). The latter is based on the Chou–Talalay methodology, which has become a fundamental tool in studying drug interactions in a quantitative and qualitative manner. Based on two main parameters, CI (combination index) and DRI (dose reduction index), the nature and significance of drug interaction is assessed in both real and simulated data sets of the constructed “dose–response” curves. The CI value is considered a major indicator of a synergistic (CI < 1), additive (CI = 1), or antagonistic (CI > 1) drug combination, whereas the DRI indicates the fold change in the equi-inhibitory concentrations of each component in the studied cotreatment regimen. In the context of cytoprotective studies aiming to establish an antagonistic pattern of interaction, the CI values of the cytotoxic component are shifted towards infinity and the DRIs are around 0. 

In our combination studies, we used a fixed 5:1 ratio of the oxidative and cytoprotective agents (H_2_O_2_:PE and H_2_O_2_:DiCQA), and for reasons of clarity and comprehensibility, we focused on the interpretation of the generated parameters at the actual treatment concentrations (Figure 2 and Figure 3, Table 2 and Table 3).

Figure 2 and Table 2 present the summarized Comusyn^®^ report from the conducted cytoprotection study with the *P. purpurea* extract (PE:H_2_O_2_ = 1:5). As evident from Figure 2, the stimulative effect of PE on neuronal cell proliferation caused a downward shift in the “dose–response” curve of the tested combination, indicating a significantly smaller affected fraction (Fa) of the cell population at the same exposure levels of the toxicant. According to Table 2, neuroprotection was most distinct at the second treatment concentration, where cell injury induced by 1000 µg/mL H_2_O_2_ was effectively attenuated by the PE pretreatment. The calculated DRI parameter for this experimental point (0.24193) indicates a nearly 4-fold reduction in the cytotoxic potential of the oxidant and a strongly antagonistic interaction (infinitely high CI). A nearly indifferent effect of the PE on cell survival was only observed in the 250:50 µg/mL treatment group, as reflected by the calculated indicators, tending to 1. At the other three experimental points, antagonistic effects were found to be similar in strength, with a twofold increase in equi-inhibitory concentrations of the damaging agent (DRIs ≈ 0.5).

The results of the cytoprotective study with the H_2_O_2_:DiCQA combination pair (Figure 3, Table 3) are even more conclusive, showing a dramatic revitalizing effect of the isolated polyphenol DiCQA on injured cell populations in a highly dose-dependent manner. Accordingly, Figure 3 shows a severe decrease in the slope of the “dose–response” curve for H_2_O_2_ under DiCQA pretreatment conditions due to the strong ameliorating effect of the phytochemical on H_2_O_2_-induced oxidative damage. Similarly to the PE combination study, the greatest improvement was seen in the cellular response at the second concentration pair, where the calculated index of dose reduction suggests a 56.7-fold decrease in H_2_O_2_-induced neurotoxicity. This antagonistic pattern is pronounced across the entire treatment range (infinitely high CI values) but gradually subsides towards the lower concentration pairs, reaching a point of non-inferiority at the final one (estimated DRI and CI close to 1).

#### 2.1.3. Results of Proteome Profiling

As discussed, oxidative stress is a key feature of H_2_O_2_-induced cytotoxicity, in view of which we conducted a pharmacodynamic investigation on the cellular response mechanisms and coping capacity of SH-SY5Y cells both in the presence and absence of the phytoprotective agents PE and DiCQA. Using proteome analysis, changes in the expression levels of related enzymes and of signaling and regulatory proteins were tracked and interpreted in the context of protein function (Figure 4).

The results obtained from the conducted immunoassays indicate the presence of complex modulating properties of the tested natural protectants on programmed cell death, inflammation, and oxidative damage-related adaptive processes. For instance, levels of the antiapoptotic bcl family member bcl-2 (1) were slightly elevated in both pretreated samples compared with untreated and unprotected control samples, exerting a stimulatory effect on cell proliferation. Regarding all other changes in protein expression and activity (2, 3, 4, 6, 7, 8), an interesting general trend was observed, where unprotected cells responded to oxidative damage with a varying degree of protein induction that appears to be aborted and/or reversed in the PE and DiCQA pretreated groups. In cytochrome c levels, another indicator of mitochondrial apoptotic events, a slight up-regulation was observed in the neuronal injury model, which was more effectively suppressed in the DiCQA pretreatment regimen. Although no changes were observed in the activation status of the cell cycle regulator p21 (5), the cyclin-CDK inhibitor p27 was prominently induced under H_2_O_2_ exposure. Consistent with the general trend, its levels returned to control values in response to PE preincubation and were even more effectively reduced in the DiCQA cotreatment regimen (about 60%). In the H_2_O_2_-damaged neuronal population, oxidative stress led to a 30% elevation in the levels of cited-2 (2), an essential differentiating factor, which was effectively averted in both the PE- and DiCQA-pretreated samples.

Changes in a number of proteins with direct relevance to oxidative stress were also observed. Hypoxia-inducible factor (HIF-1α, 6) and indoleamine 2, 3-dioxygenase (IDO, 7) demonstrated similar expression profiles, obeying the general trend across samples. Once again, a slightly more adequate response to hypoxia provided the polyphenolic DiCQA, reducing HIF-1α below control values. The variations in IDO levels between groups were much more subtle; however, both phytoprotectants produced a mild decrease, thus alleviating the neuroinflammatory, oxidative, and apoptotic effects of the enzyme. Finally, overexpression of the inducible COX-2 in response to the noxious inflammatory stimulus in the H_2_O_2_-injured population was effectively down-regulated (by about 30%) to a lower level than the one measured in the naive control group.

### 2.2. UHPLC-DAD Analysis

A UHPLC-DAD method for the quantitative determination of the main compounds in *P. purpura* extract (PE) was developed. A total of 14 analytes, including hydroxycinnamic acids (caffeic (3) and *p*-coumaric (7) acids), 7 acylquinic acids (2, 5, 5, 6, 10, 11, 12, and 14), 2 caffeoyltartaric acids (caftaric (1) and cichoric (8) acids), and the flavone luteolin (13) and its *O*-glycoside (9) were determined in PE. Based on the UV spectra and previous LC-HRMS data compounds, 4, 5, 6, 11, 12, and 14 were assigned as acylquinic acids (25). The UHPLC-DAD chromatogram of PE is presented in Figure 5. The content of the assayed compounds is revealed in Table 4. 3,5-diCQA (10) was the major compound in PE, followed by cichoric acid (8), chlorogenic acid (2), and AQA6 (14).

## 3. Materials and Methods

### 3.1. Plant Material and Sample Extraction

The collection of plant material and sample extraction were previously described by Mihaylova et al. [25]. *P. purpurea* aerial parts were collected at Vitosha Mt., Bulgaria, in July 2022. A voucher specimen was deposited at the Herbarium Academiae Scientiarum Bulgar-iae (SOM 177 803). Air-dried material was extracted twice with 80% MeOH (1:20 *w*/*v*) by sonication (80 kHz, ultra-sound bath Biobase UC-20C) for 15 min at room temperature. The extracts were concentrated in vacuo and subsequently lyophilized. The crude *P. purpurea* extract (PE) was subjected to further pharmacological tests and UHPLC-DAD analyses.

#### Chemicals

Acetonitrile (hypergrade for LC-MS), formic acid (for LC-MS), and methanol (analytical grade) were purchased from Chromasolv (Sofia, Bulgaria). The reference standards used for compound identification were obtained from Extrasynthese (Genay, France) for *p*-coumaric, luteolin 7-*O*-glucoside, and luteolin. Chlorogenic, caffeic, caftaric, cichoric, and 3,5-dicaffeoylquinic acid were supplied from Phytolab (Vestenbergsgreuth, Germany). The stock solution H_2_O_2_ (30% *w*/*v* hydrogen peroxide (extra pure, suitable for analysis) was purchased from Fisher Chemical™ (Waltham, MA, USA). 

### 3.2. In Vitro Cytotoxicity Assays

#### 3.2.1. Cell Lines and Culture Conditions

The neuroprotective activity of *Prenanthes extract* and DiCQA was assessed against H_2_O_2_-induced cell stress in a human neuroblastoma cell line (SH-SY5Y) purchased from the German Collection of Microorganisms and Cell Cultures (DSMZ GmbH, Braunschweig, Germany). Cell cultures were cultivated in the growth medium RPMI 1640 supplemented with 10% fetal bovine serum (FBS) and 5% L-glutamine and incubated under standard conditions of 37 °C and 5% humidified CO_2_ atmosphere.

#### 3.2.2. MTT Cell Viability Assay

The cell viability of SH-SY5Y cells following a 30 min exposure to the neurotoxic H_2_O_2_ was assessed in the following experimental settings: unprotected control samples and cell cultures pre-incubated for 24 h with PE and 3,5-DiCQA, intended as phytoprotectants. The degree of H_2_O_2_-induced cytotoxicity in each experimental scenario was evaluated against the untreated control group using a validated methodology for assessing cell viability known as the Mosmann MTT method. Exponential-phase cells were harvested and seeded (100 μL/well) in 96-well plates at the appropriate density (1.5 × 10^5^). On the following day, pretreatment for 24 h was conducted with PE and 3,5-Dicaffeoylquinic acid in the concentration range 400–25 µg/mL. Both the protected and unprotected treatment groups were further exposed to 5-fold serial dilutions of the oxidizing agent H_2_O_2_ (2000–125 µg/mL) for 30 min. Filter-sterilized MTT substrate solution (5 mg/mL in PBS) was added to each well of the culture plate. A further –4 h incubation allowed for the formation of purple insoluble formazan crystals. The latter were dissolved in isopropyl alcohol solution containing 5% formic acid prior to absorbance measurement at 550 nm. Collected absorbance values were blanked against an MTT and isopropanol solution and normalized to the mean value of the untreated control (100% cell viability).

#### 3.2.3. Chou–Talalay Method

The occurrence and quantitative determination of the neuroprotective effects of both natural products (PE and 3,5-DiCQA) were established using Compusyn^®^ software based on the Chou–Talalay methodology. The automated analysis was performed based on the priorly derived “dose–response” data for each experimental point, according to the protocol described in the previous section. The antagonistic activity of the tested PE and the natural compound DiCQA against the oxidizing agent H_2_O_2_ was measured at a fixed (constant) 1:5 ratio between their treatment concentrations at all exposure levels with a 24 h preincubation with the phytoprotectants. Their enhancing effect on cell survival and tolerability to the harmful stimulus was evaluated and interpreted for each experimental data set (actual treatment concentrations) and the nature of the studied drug interactions was determined based on the automated analysis in Compusyn software and the generated combination (CI) and drug reduction (DRI) indices. The CI provides a quantitative determination of synergistic (CI < 1), additive (CI = 1), and antagonistic (CI > 1) drug behavior in fixed- or varying-ratio combinations. Similarly, the DRI (dose reduction index)–Fa plot indicates the fold change in the equi-effective concentrations of a single drug when used in combination (DRI > 1 for synergistic interactions and 0 > DRI < 1 for antagonistic behavior).

### 3.3. Proteomic Analysis

A series of immunoassay experiments were performed to monitor the molecular aspects of oxidative changes and coping capacity of H_2_O_2_-injured neuronal cells subjected or not to a phytoprotective pretreatment with Prenanthes extract and one of its major constituents, DiCQA. Changes in the expression levels and activation status of proteins related to oxidative stress in response to H_2_O_2_ exposure (500 µg/mL) were tracked and analyzed in a comparative manner to untreated control and pretreated samples (100 µg/mL) in membrane-based sandwich immunoassays conducted according to the manufacturer’s instructions (Proteome Profiler Human Cell Stress Array Kit, R&D Systems). The doses used in the cotreatment regimen were selected in respect to the cytotoxicity data obtained from the MTT test and the quantitative Compusyn^®^ analysis (version 1.0). The proteins were visualized using a digital imaging system (Azure Biosystems C600) and densitometric analysis of the array spots was conducted using ImageJ^®^ software (version number 1.54). The most prominent changes in spot signals were expressed graphically relative to untreated control and interpreted in a comparative manner to the pretreated samples.

### 3.4. UHPLC-DAD Analysis

UHPLC-DAD analyses were carried out on a Thermo Scientific Dionex UltiMate 3000 analytical system equipped with a Dionex UltiMate 3000 RS Pump (LPG-3400RS), Dionex UltiMate 3000 RS Autosampler (WPS-3000TRS), Dionex UltiMate 3000 RS Column Compartment (TCC3000RS), and Dionex UltiMate 3000 Diode Array Detector (DAD-3000) and provided by Thermo Fisher Scientific (Germering, Germany). Separation and quantitative analysis were achieved as previously described by Mihaylova et al., 2024 [31].

## 4. Conclusions

Natural sources of compounds with antioxidant properties have been a focal point of research as either preventive or mitigating measures to tackle oxidative stress and related pathologies. The present study makes a valuable contribution to the existing body of research on polyphenolic compounds and their sources as antioxidative agents. Accordingly, a 24 h preincubation of neuronal cells with the studied *P. purpurea* extract served to increase the half-inhibitory concentration of the H_2_O_2_ toxicant by about 4 times. In the studied oxidative stress model, 3,5-DiCQA exerted an even more pronounced effect on cell survival and was able to neutralize peroxide toxicity by tens of times. However, the cytoprotective responses in both pretreatment regimens followed a common general trend, where PE and 3,5-DiCQA both managed to reverse the observed induction in protein expression in the unprotected control group. The presented experimental data reconfirm the cytoprotective properties of the studied PE extract and are a credible testimony to its antioxidant action, strongly characteristic of polyphenolic compounds.

## Figures and Tables

**Figure 1 ijms-25-09805-f001:**
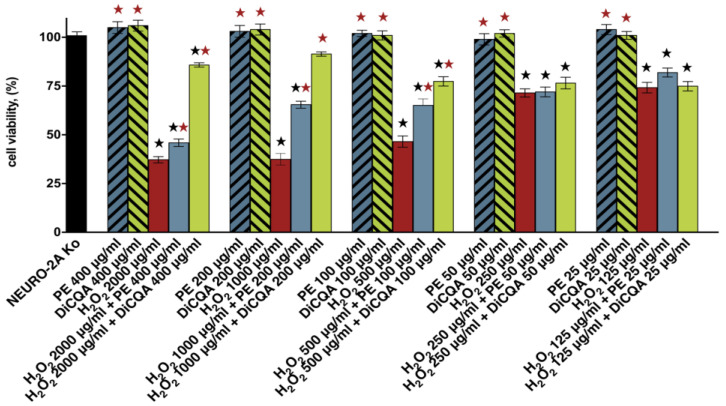
Cell viability of SH-SY5Y cells following 24 h exposure to various concentrations of H_2_O_2_: unprotected control group (red) and groups pretreated for 24 h with serial dilutions of *Prenanthes purpurea extract* (PE, in blue) and DiCQA (green) in a fixed 5:1 dose ratio. Blue and green bars in fill pattern represent the sole effects of PE and DiCQA monotreatments, respectively, on cell viability. All experiments were run in triplicate and data are expressed as mean ± SD. Statistical significance of the data was assessed via a two-way ANOVA (black ★ *p* ≤ 0.01 vs. untreated control; red ★
*p* ≤ 0.01 vs. unprotected control group).

**Figure 2 ijms-25-09805-f002:**
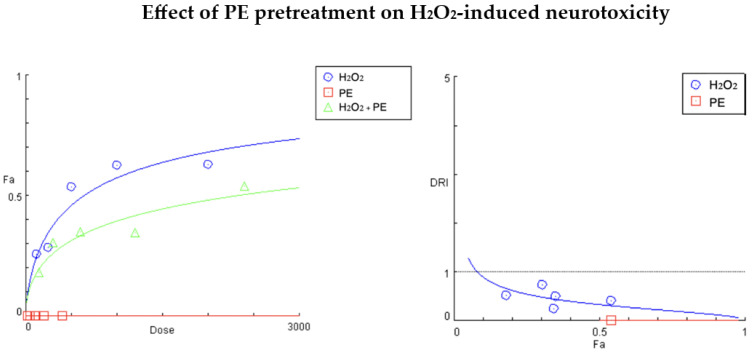
“Dose–response” curves (**left**) and DRI/Fa plot (**right**) for the experimental data points of the studied combination PE + H_2_O_2_. Fa (affected fraction) indicates the percentage of cell growth inhibition expressed as a part of 1, whereas DRI (the dose reduction index) defines the fold change in the equi-effective concentrations of H_2_O_2_ when used in a pretreatment regimen (DRI > 1 for synergistic interactions and 0 > DRI < 1 for antagonistic behavior). The DRI/Fa plot gives the estimated DRI values for H_2_O_2_ at all five Fa values (degrees of growth inhibition at the tested treatment concentrations).

**Figure 3 ijms-25-09805-f003:**
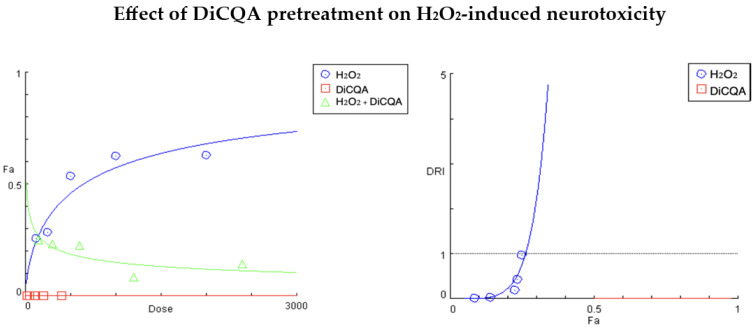
“Dose–response” curves and DRI-Fa plot for the experimental data points of the studied combination DiCQA + H_2_O_2_. Fa (fraction affected) indicates the percentage of cell growth inhibition expressed as a part of 1, whereas DRI (dose reduction index) defines the fold change in the equi-effective concentrations of H_2_O_2_ when used in a pretreatment regimen (DRI > 1 for synergistic interactions and 0 > DRI < 1 for antagonistic behavior). The DRI/Fa plot gives the estimated DRI values for H_2_O_2_ at all five Fa values (degrees of growth inhibition at the tested treatment concentrations).

**Figure 4 ijms-25-09805-f004:**
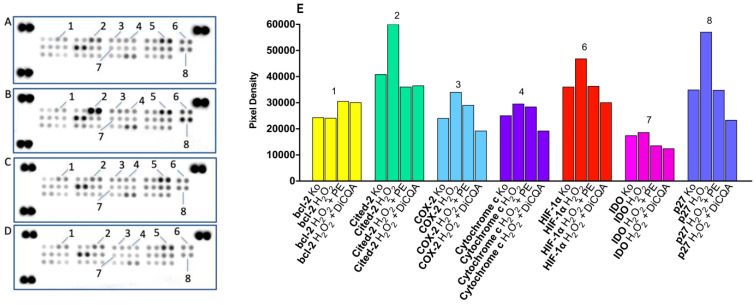
Changes in expression levels of oxidative cell stress-related proteins in SH-SY5Y cells exposed for 30 min to H_2_O_2_ in unprotected (**B**) and PA (**C**) and DiCQA (**D**) pretreatment regimens, as compared to untreated control (**A**). Combination treatments were conducted in the same 5:1 dose ratio (500 H_2_O_2_: 100 µg/mL phytoprotectant), with the phytoprotection preceding the noxious stimuli by 24 h. After a total 48 h incubation, a human proteome profiler immunoassay was performed according to the manufacturer’s instructions. Further densitometric analysis of the array spots was conducted using ImageJ software (version number 1.54) and the most prominent changes in the proteome were expressed graphically (**E**). Legend: 1—bcl-2; 2—Cited-2; 3—COX-2; 4—Cytochrome c; 5—p21; 6—HIF-1α; 7—IDO; 8—p27.

**Figure 5 ijms-25-09805-f005:**
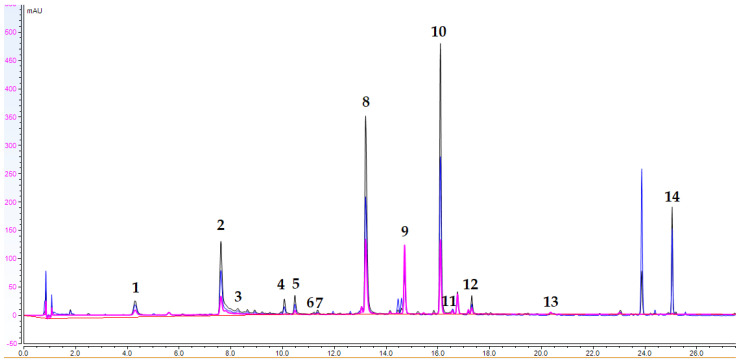
UHPLC-DAD chromatograms of PE; wavelengths: 360 nm, 310 nm, 280 nm (peak assignments are listed in Table 4).

**Table 1 ijms-25-09805-t001:** In vitro cytotoxicity of H_2_O_2_ [IC_50_, µg/mL ± SD] against SH-SY5Y cells when used alone or in a pretreatment regimen with PE and DiCQA.

Cell Line/Treatment Group	H_2_O_2_	H_2_O_2_ + PE	H_2_O_2_ +DiCQA
SH-SY5Y ^1^	583.5 ± 18.4	>2000	>7000

^1^ Human neuroblastoma cell line.

**Table 2 ijms-25-09805-t002:** Dose reduction indices (DRIs) and combination indices (CIs) estimated for the experimental data points of the studied combination PE + H_2_O_2_. CI provides a quantitative determination of synergistic (CI < 1), additive (CI = 1), or antagonistic (CI > 1) drug behavior at the studied fixed ratio combination. The DRI indicates the fold change in the equi-effective concentrations of H_2_O_2_ when used in a PE pretreatment regimen (DRI > 1 for synergistic interactions and 0 > DRI < 1 for antagonistic behavior).

Fa (H_2_O_2_ Alone)	Fa’ (Combo)	Dose H_2_O_2_	Dose PE	DRI H_2_O_2_	CI Value
0.628	0.540	2000.0	400.0	**0.40569**	**Infinit**
0.625	0.345	1000.0	200.0	**0.24193**	**Infinit**
0.535	0.349	500.0	100.0	**0.49694**	**Infinit**
0.285	0.270	250.0	50.0	**0.93989**	**→ 1**
0.258	0.18	125.0	25.0	**0.51623**	**Infinit**

**Table 3 ijms-25-09805-t003:** Dose reduction indices (DRIs) and combination indices (CIs) estimated for the experimental data points of the studied combination DiCQA + H_2_O_2_. CI provides a quantitative determination of synergistic (CI < 1), additive (CI = 1), and antagonistic (CI > 1) drug behavior at the studied fixed ratio combination. The DRI indicates the fold change in the equi-effective concentrations of H_2_O_2_ when used in a DiCQA pretreatment regimen (DRI > 1 for synergistic interactions and 0 > DRI < 1 for antagonistic behavior).

Fa (H_2_O_2_ Alone)	Fa’ (Combo)	Dose H_2_O_2_	Dose DiCQA	DRI H_2_O_2_	CI Value
0.628	0.142	2000.0	400.0	0.02106	Infinit
0.625	0.085	1000.0	200.0	0.01761	Infinit
0.535	0.225	500.0	100.0	0.19684	Infinit
0.285	0.234	250.0	50.0	0.42513	Infinit
0.258	0.251	125.0	25.0	0.97783	1.09452

**Table 4 ijms-25-09805-t004:** Content (μg/mg dry extract) of compounds assayed in PE.

№	Analyte	t_R_	Content (μg/mg de)
1.	caftaric acid	4.30	4.78 ± 0.13
2.	chlorogenic acid	7.61	11.80 ± 0.03
3.	caffeic acid	8.27	1.73 ± 0.10
4.	AQA1	10.07	1.93 ± 0.04
5.	AQA2	10.47	2.49 ± 0.10
6.	AQA3	11.12	0.090 ± 0.008
7.	*p*-coumaric acid	11.35	0.26 ± 0.01
8.	cichoric acid	13.21	26.63 ± 1.45
9.	luteolin 7-*O*-glucoside	14.71	2.33 ± 0.27
10.	3,5-diCQA	16.09	30.30 ± 3.24
11.	AQA4	16.76	1.68 ± 0.33
12.	AQA5	17.31	1.73 ± 0.10
13.	luteolin	20.36	0.100 ± 0.002
14.	AQA6	25.05	7.60 ± 0.67

## Data Availability

The raw data supporting the conclusions of this article will be made available by the authors on request.

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
