# Peer review of "Prenanthes purpurea and 3,5-DiCQA Alleviate Cellular Stress in H2O2-Induced Neurotoxicity: An In Vitro Comparative Study"

_ijms, 2024, doi:10.3390/ijms25189805_

Round 1

Reviewer 1 Report

Comments and Suggestions for Authors

The authors presented results of a study of antioxidative potential of extract obtained from the plant Prenanthes purpurea and its main component DiCQA on neuroblastoma cells. The study is well designed and well performed and it may represent the added value to the knowledge regarding natural antioxidans. Still, there are some flaws that should be corrected. Results are not clearly presented and, as mentioned in remarks to the authors, Some tables and figures are missing results obtained in isolated treatments with PE or DiCQA or hydrogen peroxide (not pretreatment followed by exposure to peroxide). Statistical analysis is missing. Without comparing treatments with PE or DiCQA only it could not be concluded whether extract or component itself exhibit cytotoxic effect or affect the gene expression. These flaws should be solved before the manuscript may be further considered for publication.

Please, write latin wordings (in vitro, in vivo, etc) in italic throughout entire manuscript.

Introduction

Lines 29-30: Please replace the wording „mitochondrial mutation“ with more precise wording because mitochondria cannot mutate. Their DNA may, and mitochondria's membrain may be damaged leading to impaired protein gradient and release of cations into cytoplasm.

Line 53: It may be a subject of a debate whether brain or liver is a major ROS producer. Due to metabolic and other physiological functions of liver, it produces significant amount of ROS.

Line 58: NO is a signaling molecule, but reactive nitrogen species (RNS) that damage macromolecules are derived from it. These are nitrite radical, and peroxinitrite radical.

Methodology

Line 302: Please, replace word „neurotoxin“ with „ROS“. Since H2O2 and ROS are not organ specific I would prefere not to categorize it as neurotoxin.

Lines 302-303: It is not clear what the authors ment exactly by „was measured 302 in a fixed 1:5 combination ratio“. Please do introduce clarification what this ratio is reffering to.

Line 318: In the sentences „Changes in oxidative stress related proteins...“ it is not clear what kind of changes was monitored – change in the ratio between them, change in the level of them, change in their structure...please, specify what changes did you monitor.

Results and Discussion

Obtained results shoulbe statistically analyzed. Only by statistically comparing results obtained in pretreatment to results obtained without it.

Statistical analysis is crucial to be able to relevantly interpret and discuss results of the study. It is mandatory.

Mass-concentration of the hydrogen peroxide that was used in treatment was presented in the results. Please, provide data on the stock solution of the hydrogen peroxide (percentage) and how the mass-concentration was calculated (µg/mL).

What represent figures in the Table 1 – live or dead cellls, are those the total numbers of cells, some indexes, recalculations...? It should be clarified in the table header.

Either, result for the peroxide treatment without pretreatment, and for treatments with either PE or DiCQA without peroxide treatment should be presented in each table and chart.

Table 1: It is not clear why mass-concentration of peroxide is presented as the mean value and SD? What was the exact concentration that was applied in the treatment?

Line 126: This conclusion cannot be reached without presenting results with PE and DiCQA treatments without the post-treatment with peroxide.

Author Response

Response to Reviewer 1 Comments

Reviewer #1: The authors presented results of a study of antioxidative potential of extract obtained from the plant Prenanthes purpurea and its main component DiCQA on neuroblastoma cells. The study is well designed and well performed and it may represent the added value to the knowledge regarding natural antioxidans. Still, there are some flaws that should be corrected. Results are not clearly presented and, as mentioned in remarks to the authors, Some tables and figures are missing results obtained in isolated treatments with PE or DiCQA or hydrogen peroxide (not pretreatment followed by exposure to peroxide). Statistical analysis is missing. Without comparing treatments with PE or DiCQA only it could not be concluded whether extract or component itself exhibit cytotoxic effect or affect the gene expression. These flaws should be solved before the manuscript may be further considered for publication.

  1. Reviewer #1: Please, write latin wordings (in vitro, in vivo, etc) in italic throughout entire manuscript.

Response: Thank you for your note, the font has been corrected accordingly.

Introduction

  1. Reviewer #1: Lines 29-30: Please replace the wording „mitochondrial mutation“ with more precise wording because mitochondria cannot mutate. Their DNA may, and mitochondria's membrain may be damaged leading to impaired protein gradient and release of cations into cytoplasm.

Response: Thank you for your note. Certainly, by the phrase in question, the authors are referring to mitochondrial DNA mutations. It has been corrected and clarified in the text.

  1. Reviewer #1: Line 53: It may be a subject of a debate whether brain or liver is a major ROS producer. Due to metabolic and other physiological functions of liver, it produces significant amount of ROS.

Response: Thank you for your note. The authors by no means underestimate the central role of the liver as a site for ROS generation given its redox activity.  We are simply highlighting the robust production of ROS in brain tissues, which is highly involved in neurodegenerative processes. 

  1. Reviewer #1: Line 58: NO is a signaling molecule, but reactive nitrogen species (RNS) that damage macromolecules are derived from it. These are nitrite radical, and peroxinitrite radical. 

Response: Thank you for your comment. Indeed, NO is ubiquitously produced and has multimodal effects. We have further clarified the sentence, referring to the secondary RNS, peroxynitrite (ONOO−), which is suggested to participate in deleterious inflammatory and degenerative processes.

Methodology

  1. Reviewer #1: Line 302: Please, replace word „neurotoxin“ with „ROS“. Since H2O2 and ROS are not organ specific I would prefere not to categorize it as neurotoxin.

Response: Thank you for your observation. Without question, the two terms are not mutually interchangeable. Referring to H2O2 as a neurotoxin is solely on the basis that it is our agent of choice for inducing neurotoxicity. The term has been replaced according to your recommendation.

  1. Reviewer #1: Lines 302-303: It is not clear what the authors ment exactly by „was measured 302 in a fixed 1:5 combination ratio“. Please do introduce clarification what this ratio is reffering to.

Response: Thank you for your note. The term “fixed combination ratio” is fundamental and widely used in the Chou-Talalay methodology, the protocol of which we describe in this section. It defines the maintenance of a constant ratio (in this case 1:5) between the concentrations of the two agents (PE/3,5-DiCQA and H2O2) at all treatment points, as opposed to other treatment regimens with varying concentration ratios. A further clarification has been provided in the text.

  1. Reviewer #1: Line 318: In the sentences „Changes in oxidative stress related proteins...“ it is not clear what kind of changes was monitored – change in the ratio between them, change in the level of them, change in their structure...please, specify what changes did you monitor.

Response: Thank you for your note.  Proteomic analysis tracks changes in the expression levels and/or activation status of the measured proteins. A further clarification has been provided in the text.

Results and Discussion

  1. Reviewer #1: Obtained results shoulbe statistically analyzed. Only by statistically comparing results obtained in pretreatment to results obtained without it. Statistical analysis is crucial to be able to relevantly interpret and discuss results of the study. It is mandatory.

Response: Thank you for your note. Statistical analysis has been provided in the revised version.

  1. Reviewer #1: Mass-concentration of the hydrogen peroxide that was used in treatment was presented in the results. Please, provide data on the stock solution of the hydrogen peroxide (percentage) and how the mass-concentration was calculated (µg/mL).

Response: Thank you for the question. A 30% (w/v) hydrogen peroxide (extra pure, suitable for analysis, Fisher Chemical™) was used as a stock solution. The concentration of this stock solution is 30 g/100 mL, equivalent to 300 000 μg/mL. The first treatment concentration we used was 2000 μg/mL, which we achieved with a 150-fold dilution in a cell culture medium. Subsequent falling treatment concentrations were derived with standard serial dilutions.

  1. Reviewer #1: What represent figures in the Table 1 – live or dead cellls, are those the total numbers of cells, some indexes, recalculations...? It should be clarified in the table header.

Response: Table 1 summarizes the estimated in vitro cytotoxicity of H2О2 in the different treatment regimens with and without pre-incubation with PE and 3,5-DiCQA. The values presented are the half-inhibitory (IC50) concentrations of the agent. It has been further clarified in the caption of Table 1.

  1. Reviewer #1: Either, result for the peroxide treatment without pretreatment, and for treatments with either PE or DiCQA without peroxide treatment should be presented in each table and chart.

Response: Results for the peroxide treatment without pretreatment ARE presented in all figures and tables, as the cytoprotective effect of the tested substances was measured against it. Monotreatments with PE and 3,5-DiCQA in the same concentration range had also been conducted, which proved to be devoid of any cytotoxic activity of their own. To avoid unnecessary complications and overloading, these cytotoxicity data were originally excluded from Figure 1. However, in response to your request, we provide an updated column graph including the sole effects of both protective agents in the tested concentration range. As compared to untreated control, no statistically significant effect of PE and 3,5-DiCQA was found on cell viability, which has been pointed out in the revised manuscript.

  1. Reviewer #1: Table 1: It is not clear why mass-concentration of peroxide is presented as the mean value and SD? What was the exact concentration that was applied in the treatment?

Response: Thank you for your note. As previously mentioned, Table 1 summarizes the estimated in vitro cytotoxicity of H2О2 in the different treatment regimens in terms of IC50 (which is presented as a mean value with SD). In the submitted manuscript, "IC50" next to the noted measurement unit and SD was accidentally left out of the caption of Table 1. The mistake has been corrected. 

  1. Line 126: This conclusion cannot be reached without presenting results with PE and DiCQA treatments without the post-treatment with peroxide.

Response: As previously mentioned, monotreatments with PE and 3,5-DiCQA in the same concentration range did not produce a statistically significant effect on cell viability. The conclusion stated in Line 126 is based on the results with subsequent H2O2-induced cell injury and indicates the observed “on demand” cytoprotective activity of PE and 3,5-DiCQA, evoked by stressful stimuli.

Reviewer 2 Report

Comments and Suggestions for Authors

Here are some comments and suggestions for the article:

1. Did the authors characterize their tested P. purpurea extract? Why was the compound 3,5-DiCQA chosen as the positive control instead of other compounds? Is a similar polyphenolic compound or the same one present in the plant extract?

2. Statistical analysis done for cell viability assay? Details missing in the figure legend. Number of repeats?

3. Figure 2 should be redrawn for better presentation. It is difficult to see with the small font labels and thin lines. The green color is also barely visible against the white background. Are there any statistical analysis performed for the study 'Effect of PE pretreatment on H2O2-induced neurotoxicity'? The figure legend should include more detailed information and also the cell lines used.

This is the same case for figure 3 and its figure legends.

4. Figure 4 should be redone. It is difficult for the reader to understand and the font labels are too small. 

5. Since the authors did the UHPLC-DAD method to quantitatively determine the main components of PE, is the quantity of each of the compound components consistent for each batch of extract? Which of these compounds is responsible for the mentioned 'alleviating neurotoxicity effect'? Is PE toxic to normal neurocells? What is the effective concentration?

6. Were the plant samples collected verified? Can the authors provide the certification of verification?

7. The study focuses on the PE and also the specific polyphenolic compound 3, 5-DiCQA. It is not clear in the article how are they related to each other. Are they from the same plant? If not why are they placed in the same study? Do they have synergistic effects? 

8. The aims and objectives of the study could be stated with greater clarity and the significance of this study should also be highlighted. 

9. For the proteomic analysis study mentioned. Did the authors perform supporting studies to verify the affected proteomic profiles from their array results? This is important to support their claims and statements. How many repeats were performed for this study and what is the sample size? Western blot or pcr studies are rendered to support their proteomic analysis results.

Comments on the Quality of English Language

English language can be comprehended but editing is required for improved coherency and clarity. There are also many typo mistakes and grammatical errors that needs to be corrected.

Author Response

Response to Reviewer #2 Comments

Here are some comments and suggestions for the article:

  1. Reviewer #2: Did the authors characterize their tested P. purpurea extract? Why was the compound 3,5-DiCQA chosen as the positive control instead of other compounds? Is a similar polyphenolic compound or the same one present in the plant extract?

Response: Thank you for your questions. In the Introduction section, it was specified that our previous study on the P. purpurea extract revealed the presence of more than 100 specialized metabolites, dereplicated/annotated by ultra-high-performance liquid chromatography - high-resolution mass spectrometry. The occurrence of acylquinnic acids (chlorogenic, 3,4-dicaffeoylquinic, 3,5-dicaffeoylquinic) was evidenced. Moreover, 3,5-DiCQA was found to be the major compound (30.30±3.24 μg/mg de) in the tested extract (See Table 4, Figure 5), on which basis 3,5-DiCQA was chosen as a positive control.

  1. Reviewer #2: Statistical analysis done for cell viability assay? Details missing in the figure legend. Number of repeats?

Response: Thank you for your note. Statistical analysis and a number of repeats have been provided in the revised manuscript.  

  1. Reviewer #2: Figure 2 should be redrawn for better presentation. It is difficult to see with the small font labels and thin lines. The green color is also barely visible against the white background. Are there any statistical analysis performed for the study 'Effect of PE pretreatment on H2O2-induced neurotoxicity'? The figure legend should include more detailed information and also the cell lines used.

This is the same case for figure 3 and its figure legends.

Response: Thank you for your recommendation. Unfortunately, Figures 2 and 3, presenting the “dose-response” curves and the DRI/Fa plots for both pretreatment regimens are generated in the original CompusynⓇ software which does not support graphical changes, refinements, and style modifications. The provided graphs are authentic and specific to the Chou-Talalay methodology. However, we have improved and further clarified their captions.

  1. Reviewer #2: Figure 4 should be redone. It is difficult for the reader to understand and the font labels are too small. 

Response: Thank you for the recommendation. Figure 4 has been improved and the font size of the labels has been increased.

  1. Reviewer #2: Since the authors did the UHPLC-DAD method to quantitatively determine the main components of PE, is the quantity of each of the compound components consistent for each batch of extract? Which of these compounds is responsible for the mentioned 'alleviating neurotoxicity effect'? Is PE toxic to normal neurocells? What is the effective concentration?

Response: Thank you for your comment. All experiments are conducted with the same batch of plant extract, guaranteeing the same PE content and phytochemical profile in all studies. The PE was in no case toxic to neurocells, on the contrary, due to its high polyphenolic content it exerted an alleviating effect on neurotoxicity (as stated). 

  1. Reviewer #2: Were the plant samples collected verified? Can the authors provide the certification of verification?

Response: Thank you for your note. A voucher specimen of the plant sample was provided (See 3. Materials and Methods, 3.1. Plant Material and sample extraction).

  1. Reviewer #2: The study focuses on the PE and also the specific polyphenolic compound 3, 5-DiCQA. It is not clear in the article how are they related to each other. Are they from the same plant? If not why are they placed in the same study? Do they have synergistic effects? 

Response: Thank you for the comment. We corrected the Abstract according to your comments (See Abstract). We already have clarified the relation between 3,5-DiCQA and P. purpurea extract. 3,5-DiCQA was found to be the major compound (30.30±3.24 μg/mg de) in the tested extract (See Table 4, Figure 5) and was commercially provided by Phytolab (Vestenbergsgreuth, Germany) (See 3.1.1. Chemicals). In our study, PE and 3,5-DiCQA are used alternatively to measure and compare their cytoprotective capacity against H2O2-induced cell injury. The possible synergistic effects between them are not an objective of the present work. Furthermore, evaluating the synergistic effects between an extract (PE, which contains more than 100 specialized metabolites) and one of its components (3,5-DiCQA) would be unreasonable.

  1. Reviewer #2: The aims and objectives of the study could be stated with greater clarity and the significance of this study should also be highlighted. 

Response: We have introduced alterations in the Introduction to better communicate the aims and objectives of this study.

  1. Reviewer #2: For the proteomic analysis study mentioned. Did the authors perform supporting studies to verify the affected proteomic profiles from their array results? This is important to support their claims and statements. How many repeats were performed for this study and what is the sample size? Western blot or pcr studies are rendered to support their proteomic analysis results.

Response: Thank you for your questions. As stated in the manuscript, the proteomic analysis was conducted according to the manufacturer's instructions (Proteome Profiler Human Cell Stress Array Kit, R&D Systems), regarding sample size, sample preparation, storage, etc. Proteome analysis is a modern (albeit expensive) highly sensitive and efficient immuno-based method that allows the parallel detection and semi-quantification of a large number of proteins in a single sample. This is also its main advantage over the alternative Western blotting, where a separate sample is required to track each protein. Both methods rely on the same principle (highly specific antigen-antibody reactions) and are not only interchangeable immunological assays, but proteomic analysis provides much more extensive information on the levels and activity of functionally related proteins in different cellular pathways and processes. Results obtained from proteome analysis using a commercially available array kit do not in any case require validation by Western blotting. As intended by the manufacturer, all proteins are determined in duplicate ( array membranes presented in Figure 4, A-D).

Round 2

Reviewer 1 Report

Comments and Suggestions for Authors

Thank you for considering the remarks to improve the manuscript's quality.

Reviewer 2 Report

Comments and Suggestions for Authors

The authors have made edits to the revised manuscript however the major concern in the lack of additional experimental data to support the proteomics results is not addressed. 

Comments on the Quality of English Language

Minor editing of English required for improved coherency.